# Regulatory Effects of Thai Rice By-Product Extracts from *Oryza sativa* L. cv. Bue Bang 3 CMU and Bue Bang 4 CMU on Melanin Production, Nitric Oxide Secretion, and Steroid 5α-Reductase Inhibition

**DOI:** 10.3390/plants12030653

**Published:** 2023-02-02

**Authors:** Warintorn Ruksiriwanich, Pichchapa Linsaenkart, Chiranan Khantham, Anurak Muangsanguan, Korawan Sringarm, Pensak Jantrawut, Chanakan Prom-u-thai, Sansanee Jamjod, Supapohn Yamuangmorn, Chaiwat Arjin, Pornchai Rachtanapun, Kittisak Jantanasakulwong, Yuthana Phimolsiripol, Francisco J. Barba, Sarana Rose Sommano, Romchat Chutoprapat, Korawinwich Boonpisuttinant

**Affiliations:** 1Department of Pharmaceutical Sciences, Faculty of Pharmacy, Chiang Mai University, Chiang Mai 50200, Thailand; 2Cluster of Research and Development of Pharmaceutical and Natural Products Innovation for Human or Animal, Chiang Mai University, Chiang Mai 50200, Thailand; 3Cluster of Agro Bio-Circular-Green Industry, Faculty of Agro-Industry, Chiang Mai University, Chiang Mai 50100, Thailand; 4Department of Animal and Aquatic Sciences, Faculty of Agriculture, Chiang Mai University, Chiang Mai 50200, Thailand; 5Lanna Rice Research Center, Chiang Mai University, Chiang Mai 50200, Thailand; 6School of Agro-Industry, Faculty of Agro-Industry, Chiang Mai University, Chiang Mai 50100, Thailand; 7Department of Preventive Medicine and Public Health, Food Science, Toxicology and Forensic Medicine, Faculty of Pharmacy, University of Valencia, 46100 Valencia, Spain; 8Department of Plant and Soil Sciences, Faculty of Agriculture, Chiang Mai University, Chiang Mai 50200, Thailand; 9Department of Pharmaceutics and Industrial Pharmacy, Faculty of Pharmaceutical Sciences, Chulalongkorn University, Bangkok 10300, Thailand; 10Innovative Natural Products from Thai Wisdoms (INPTW), Faculty of Integrative Medicine, Rajamangala University of Technology Thanyaburi, Pathumthani 12130, Thailand

**Keywords:** androgenetic alopecia, Bue Bang 3 CMU, Bue Bang 4 CMU, rice by-products, hair graying, hair growth promotion, melanin production, *Oryza sativa*, 5α-reductase

## Abstract

Alopecia and gray hair are common hair abnormalities affecting physical appearance and causing psychological problems. Chemical treatments partially restore hair disorders but have distressing side effects. Bioactive plant compounds constitute promising sources of potential medicinal substances instead of chemical agents, producing high side effects. In this study, we focused on the waste of local rice cultivars: Bue Bang 3 CMU (BB3CMU) and Bue Bang 4 CMU (BB4CMU) from the north of Thailand. The rice bran oil (RBO), defatted rice bran extract (DFRB), and rice husk (H) were determined for in vitro hair revitalization in melanin production, nitric oxide (NO) secretion, and steroid 5α-reductase inhibition. The results indicated that BB4CMU-RBO with high contents of iron, zinc, and free fatty acids showed a comparable induction of melanin production on melanocytes (130.18 ± 9.13% of control) to the standard drug theophylline with no significant difference (*p >* 0.05). This promising melanin induction could be related to activating the NO secretion pathway, with the NO secretion level at 1.43 ± 0.05 µM. In addition, BB4CMU-RBO illustrated a significant inhibitory effect on both steroid 5α-reductase genes (*SRD5A*) type 1 and type 2, which relates to its primary source of tocopherols. Hence, rice bran oil from the Thai rice variety BB4CMU could be applied as a promising hair revitalizing candidate, from natural resources, to help promote hair growth and re-pigmentation effects.

## 1. Introduction

Plant-based substances have been recently developed for current global trends in the cosmeceutical industry, depending on customer behaviors and demands [1]. Furthermore, active compounds from plant resources, especially agricultural by-products, are environmentally-friendly with health benefits [2,3]. Modifying agricultural residues into value-added products is a sustainable development method for effectively using natural resources while reducing harmful environmental impacts [4]. It also complies with the Sustainable Development Goals (SDGs), such as SDG 3: Good health and well-being and SDG 12: Responsible consumption and production.

*Oryza sativa* L. (rice) is a cereal grain used for worldwide consumption. Polished rice is derived from whole grain rice and is used to prepare many kinds of food [5]. Rice bran and rice husk are major by-products generated from the rice milling process. Many studies have shown that high-value components in rice waste can be used in a variety of applications, including construction materials [6], nutritional supplements [7], and skin care products [8,9]. Similarly, we previously reported on both rice bran and rice husk extracts from the Thai rice variety, Bue Bang 3 CMU (BB3CMU), which possesses hair growth properties with high tocopherol contents [10]. Our current phytochemical studies of northern Thai rice cultivars demonstrated that oryzanol and tocopherol were profoundly enriched in BB3CMU and Bue Bang 4 CMU (BB4CMU) rice brans [11]. Taken together, both rice varieties may constitute potential sources for hair care applications. In our previous work, the extracts from rice by-products were separated into three distinct parts: non-polar rice bran oil extracts, rice bran residue extracts, and rice husk extracts. We further elucidated the phytochemical quantification of oryzanols, tocopherols, free fatty acids, and polyphenols [12].

The global market size of hair care was around USD 75.06 billion in 2020. This market is predicted to be worth USD 112.97 billion by 2028 [13]. Furthermore, hair loss treatment products and hair dyes accounted for 30–35% of the total market share among hair care products [14]. The demand for hair loss treatment products and hair colorants has rapidly increased because hair baldness and hair graying are recognized signs of aging. Visible hair problems mainly cause low self-confidence and affect general well-being as well as mental health [15,16]. Androgenetic alopecia (AGA) comprises chronic hair miniaturization affecting both male and female hair patterns. Higher levels of serum sex hormone, 5α-reductase, and androgen receptors in hair follicles are associated with premature baldness. The pathogenesis of AGA can be related to cystic acne, hirsutism, or virilization among females [17,18]. Therapeutic agents, such as topical minoxidil and antiandrogens are approved for AGA treatment [19]. The clinical evidence implicated adverse events with long-term medication. Itching, redness, and hypertrichosis on body areas are the common side effects of topical minoxidil. Moreover, sexual dysfunction, metabolic syndromes, and depression are related to an alteration in androgens [20,21].

Hair graying is visualized as a loss of pigment in the hair shaft. Melanin pigments play a crucial role in hair photoprotection, especially black eumelanin. Melanin production is regulated by various proteins, such as tyrosinase and tyrosinase-related proteins [22]. An earlier study found that inherited mutations of melanogenic proteins can cause cutaneous hypopigmentation with canities [23]. Moreover, diet and nutritional status have a considerable impact on premature hair graying. A dietary supplement is partially effective in reversing hypopigmented hair [24,25]. Hair dying is another option for covering gray hair [26]. Additionally, excipient ingredients, such as *p*-phenylenediamine, preservatives, or surfactants in hair colorants, may cause allergies and cancer [27,28].

Therefore, this study aimed to determine the biological activities of three different rice by-products (BB3CMU and BB4CMU) as botanical therapeutic agents in terms of antioxidant properties, melanin production, nitric oxide production, and any inhibitory effects on steroid 5α-reductase isoenzymes.

## 2. Results and Discussion

### 2.1. Extraction Process

Both Thai rice varieties BB3CMU and BB4CMU were successfully registered as new plant varieties through Plant Breeder’s Rights (PBR) under the Department of Agriculture, Thailand, in November 2020 and February 2022, respectively. Both rice cultivars have white non-glutinous grains with brown husks. The distinct morphological characteristics of these varieties are based on stem and leaf features. The stem diameter of BB3CMU is approximately 10.19 mm, and the leaf dimensions are 1.55 cm and 42.34 cm. In contrast, the rice stem diameter of BB4CMU is about 8.05 mm, with a larger leaf size (1.59 cm and 58.90 cm) and a hairy surface on the leaf blade and leaf sheath. According to our previous experiment, the proximate analysis indicated that rice variety BB3CMU possessed the highest crude fat content of 18.68 ± 0.11%, followed by that of BB4CMU at 18.55 ± 0.19%, among eleven rice varieties. Moreover, Wisetkomolmat et al. reported that BB3CMU and BB4CMU showed high contents of crude fiber and crude protein. For the mineral content analysis, both BB3CMU and BB4CMU demonstrated high contents of K (1.98 ± 0.00 and 2.13 ± 0.00 g/100 g sample), Mg (0.63 ± 0.00 and 0.72 ± 0.02 g/100 g sample), Ca (288.93 ± 35.96 and 310.63 ± 13.26 mg/kg sample), Fe (71.62 ± 0.74 and 83.17 ± 1.09 mg/kg sample), Mn (112.88 ± 5.76 and 136.50 ± 0.60 mg/kg sample), Zn (66.07 ± 1.73 and 73.57 ± 3.98 mg/kg sample), and Na (33.10 ± 0.78 and 39.00 ± 0.63 mg/kg sample), respectively [11].

After the screw press process, augmented with dichloromethane extraction, rice bran oils (RBO) from BB3CMU and BB4CMU are dark brown viscous oils with extraction yields of 20.12% and 19.17% *w/w* based on rice bran materials, respectively. The screw press is a favored oil extraction method without heat treatment [29]. Moreover, dichloromethane is an effective inorganic solvent for plant oil production [30]. Thus, mechanical extraction with an inorganic solvent was conducted to refine the crude oil from rice bran samples. Then, the defatted residues were macerated with an ethanol solution to obtain other semi-polar and polar compounds in defatted rice bran extracts (DFRB), comprising light brown greasy pastes. The extraction yields of both BB3CMU-DFRB and BB4CMU-DFRB were 7.67% and 7.13% *w/w* of de-oiled rice bran samples, respectively. For rice husk (H) portions, the physical attributes of both BB3CMU-H and BB4CMU-H were non-glossy crude extracts with extraction yields of 2.09% and 2.00% *w/w*, based on the dry husk mass [12].

### 2.2. In Vitro Antioxidant Activities

The antioxidant activities of *O. sativa* by-product extracts were demonstrated by the 2,2-diphenyl-1-picrylhydrazyl radical (DPPH) assay, the 2,2′-azino-bis (3-ethylbenzothiazoline-6-sulfonic acid) radical (ABTS) assay, and metal chelation. DPPH is a stable radical and can be scavenged by antioxidants in lipophilic samples, whereas the ABTS assay is used to examine both lipophilic and hydrophilic compounds [31]. Metal chelation is used to determine the chelating process with an ion-substrate complex in the system, such as ferrous-ferrozine formation [32]. The antioxidant capacity obtained from these methods correlated to biomolecules in medicinal plants, e.g., flavonoids and phenolic compounds [33,34].

The results of the present study showed that rice husk extracts possessed antioxidant properties (Table 1), which were higher than those of rice bran oil and rice bran extracts. In vitro scavenging activities using the DPPH and ABTS methods were observed for BB3CMU-H (334.70 ± 4.64 and 196.13 ± 10.09 mg TE/g extract), and BB4CMU-H (198.04 ± 2.75 and 164.10 ± 8.45 mg TE/g extract), respectively. The highest chelating activity determined using iron chelation was found for BB3CMU-H with 283.22 ± 55.35 mg EDTAE/g extract. The antioxidant effects of both BB3CMU-H and BB4CMU-H agreed with those of the high phenolic content in the rice husk portion. The bioactive compositions of the rice husk extracts were performed using the high-performance liquid chromatography (HPLC) analysis, as reported in our previous work. The rice husk extracts of *O. sativa* cv. BB3CMU and BB4CMU were found to be high in polyphenols, such as phytic acid, *p*-coumaric acid, and kaempferol [12]. In biochemical systems, phytic acid with a strong chelating activity can prevent the formation of hydroxyl radicals [35]. The previous studies showed that *p*-coumaric acid was involved in the promotion of hair growth by reducing oxidative stress [36,37]. Additionally, numerous reports revealed that kaempferol had a potent scavenging ability and regulated inflammatory responses [10,38].

### 2.3. Cell Viability

The cytotoxic effects of rice extracts were performed on RAW 264.7 (murine macro-phage), B16F10 (mouse melanoma), and DU-145 (human prostate cancer) cells. The maximum non-cytotoxic concentration (>80% cell viability) [39] of each extract was considered as the treatment concentration for subsequent experiments. The viability of RAW 264.7 cells was not altered over the concentration range of 0.0001 to 0.10 mg/mL of rice bran oils and rice husk extracts for 24 h. However, defatted rice bran extracts illustrated a higher toxicity than rice bran oil and husk fractions. The defatted rice bran extracts at 0.0001 mg/mL revealed no cytotoxicity on macrophage cells. Therefore, 0.10 mg/mL of both rice bran oils and rice husk extracts and 0.0001 mg/mL of the defatted rice bran extracts were selected for RAW 264.7 cell exposure. Moreover, all extracts at a concentration of 0.01 mg/mL showed non-cytotoxic effects on melanoma cells after 48 h of incubation. Thus, the melanin content assay would be tested with 0.01 mg/mL of all rice fractions. On prostate cancer cells, no cytotoxic effect was noticed after exposure to 0.50 mg/mL of the defatted rice bran extracts, 0.25 mg/mL of the rice bran oils, and 0.10 mg/mL of the rice husks after 24 h of incubation. These concentrations were used to further evaluate using DU-145 cells. Moreover, rice extracts were evaluated for cytotoxicity in human fibroblasts and dermal papilla cells from human hair follicles (HFDPC). The non-cytotoxic effects of extracts may not affect scalp cells. The rice extracts at 0.25 mg/mL did not show any toxicity to human fibroblasts or HFDPC (>80% cell viability of control). It could be assured that rice extracts at high concentrations might be suitable for human scalp application.

### 2.4. Effects of Oryza sativa L. Extracts on Melanin Production in Melanoma Cells

Melanin pigments in hair shafts are synthesized by the melanocytes in the hair bulb, interacting with the melanin precursors in the bulb keratinocytes [40]. Hair pigmentation is modulated by melanogenic regulatory factors, such as tyrosinase, tyrosinase-related proteins, melanocyte-stimulating hormone, and melanocortin receptor. The conversion of L-tyrosine to L-3,4-dihydroxyphenylalanine (L-DOPA) by the copper-dependent tyrosinase enzyme is the rate-limiting process in melanin biosynthesis [41,42,43]. Chelating with copper ions in the active site can inhibit this enzyme, whereas L-DOPA, superoxide anion, and nitric oxide (NO) as electron donors can activate enzyme activity [23,44]. Pigmented hair shafts are produced notably during the late anagen phase then regress and diminish in catagen to the resting stage of the hair cycle [45].

However, the lack of melanogenic melanocytes during the anagen phase is the primary cause of hair graying. The failure of melanogenic activity is a consequence of reactive oxygen species (ROS) damage along with the impairment of the antioxidative defense system due to the effects of UV radiation, smoking, and pollution [46,47]. According to the previous research, the natural antioxidant properties from *Eurotium cristatum* [48], *Melissa officinalis* [49], *Cannabis sativa* [50], and *Bixa orellana* [51] promoted melanogenesis and can be used as potential preventive and therapeutic agents for skin and hair hypopigmentation. Hence, the phenolic profile of rice husk extracts with scavenging effects may provide protection against ROS.

According to the study results, both BB3CMU and BB4CMU extracts could up-regulate melanin synthesis in B16F10 melanoma cells. Notably, the stimulating effects on melanin synthesis of BB4CMU-RBO (130.18 ± 9.13% of control) showed a comparable effect to the standard theophylline (139.73 ± 5.20% of control) with no significant difference at 0.01 mg/mL (Figure 1). Theophylline is used as a melanogenic stimulating agent affecting melanogenesis by elevating cyclic adenine monophosphate (cAMP) via the mitogen-activated protein kinase 1 (MEK 1/2) and the Wnt/β-catenin biological signaling pathways [52,53]. This result suggested that the melanogenesis stimulating ability of BB4CMU-RBO may contribute to iron, zinc, and unsaturated fatty acids. Malnutrition and vitamin deficiencies can cause or contribute to canities. Iron, zinc, copper, calcium, and cobalamin could modulate melanogenesis in hair follicles, leading to reversible graying of hair [47,54,55]. 

A previous report illustrated that iron treatment can elevate the number of maturing melanosomes, the pigmentation in melanocytes, and the expression of the melanocyte-inducing transcription factor (MITF) [56]. Interestingly, zinc is recognized as a cofactor of tyrosinase-related protein type 1 [57] and can prevent the cell apoptosis of melanocytes [58]. In addition, previous reports demonstrated that rice bran extracts with high contents of linoleic acid or silicic acid have the potential to restore the melanogenesis process [59,60,61]. Our previous study revealed the nutritional profiles of eleven rice bran varieties from the north of Thailand. The mineral components in BB4CMU surpassed other rice cultivars, especially regarding iron (83.17 ± 1.09 mg/kg rice bran sample) and zinc (73.57 ± 3.98 mg/kg rice bran sample). Moreover, BB4CMU rice bran contained a high concentration of essential unsaturated fatty acids varieties, such as oleic acid, linoleic acid, and α-linolenic acid [11], supporting the melanogenesis ability of BB4CMU-RBO.

### 2.5. Effects of Oryza sativa L. Extracts on Nitric Oxide Production in Macrophages

Nitric oxide (NO) is synthesized from the amino acid L-arginine and oxygen molecules by nitric oxide synthases (NOS). The NOS enzyme was observed in keratinocytes as well as melanocytes. NO plays an important part in the physiological system as a key signaling molecule. The role of NO may involve immune system responses, vascular function, and skin homeostatic regulation [62,63,64]. Neuronal NOS (nNOS), endothelial NOS (eNOS), and inducible NOS (iNOS) are the three isoforms of NOS [65]. The iNOS usually produces NO after lipopolysaccharide (LPS) stimulation. In the previous experiment, iNOS was expressed on the scalp epidermis, whereas nNOS and eNOS were mostly observed on epidermal melanocytes and dermal keratinocytes [66].

The measured nitrite content indicated NO production in macrophage cells [67]. As shown in Figure 2, BB4CMU-RBO displayed the greatest nitrite production (1.43 ± 0.05 µM), followed by BB4CMU-DFRB (1.10 ± 0.10 µM), which were comparable to the LPS treatment (1.70 ± 0.10 µM). Previous studies suggested that melanin synthesis could be enhanced by NO production [62,68]. After ultraviolet (UV) exposure, the intracellular level of NO is remarkably increased and then activates melanogenesis through the cyclic guanosine monophosphate (cGMP) signaling pathway [69,70]. It has been established that NO regulates soluble guanylate cyclase, which turns guanosine triphosphate (GTP) to cGMP and further elevates protein kinase G (PKG), resulting in increased expressions of pigmentation genes and melanin production [71,72,73]. Some evidence suggested that NO is associated with the complex effects in melanogenesis, including the morphological appearance of dendritic melanocytes, facilitation of melanosome aggregation, and increasing the melanin synthesis activity of melanocytes [74].

The previous articles presented that saturated fatty acids (palmitic acid and steric acid) and unsaturated fatty acids, e.g., oleic acid and linoleic acid, could induce NO production in macrophage cells, resulting in macrophage differentiation [75,76]. M1-like macrophages (antigen-presenting cells) are involved in releasing pro-inflammatory cytokines and T-cell function, which can be activated by saturated fatty acids. On the other hand, unsaturated fatty acids could influence the M2-like phenotype (alternatively activated macrophages), leading to tissue repair and anti-inflammatory cytokine secretion [75,77]. In this study, BB4CMU-RBO was rich in free fatty acids, consisting of palmitic acid, steric acid, oleic acid, and linoleic acid [11], corresponding to an increase in melanin content via NO induction. Remarkably, palmitic acid could increase tyrosinase-related proteins, resulting in inducing melanin production [78].

### 2.6. Effects of Oryza sativa L. Extracts on Gene Expression of Steroid 5α-Reductase Isoenzymes

The different stages of the hair cycle include the growth (anagen), transition (catagen), resting (telogen), and shedding (exogen) stages. Normally, a new hair matrix is formed during the anagen then regresses and undergoes apoptosis in the telogen, and the active hair follicle would regrow after the shedding step [79,80]. The hair-growth cycle is regulated by multiple signaling pathways, including the Wnt/β-catenin, sonic hedgehog (Shh), notch, angiogenesis, and transforming growth factor (TGF)-β signaling pathways. Hair development is thought to be influenced by Wnt, Shh, and vascular growth factors, whereas inflammatory cytokines, TGF-β and 5α-reductase dominate the negative aspects of hair regeneration [81,82]. In particular, the androgen-dependent pathway is prone to associate with disease-induced hair loss [83] and genetic variant-mediated alopecia [84,85].

The pathophysiology of hair loss may be complicated by complex processes. The most common hair loss disorder is AGA, which is consistent with genetic factors and androgen excess. Testosterone, the active androgen in circulation is converted to the most potent androgen, dihydrotestosterone (DHT) by 5α-reductase [21,86]. This enzyme can be divided in multiple isoforms, such as 5α-reductase type 1 (SRD5A1), type 2 (SRD5A2), and type 3 (SRD5A3). SRD5A2, located on the dermal papilla, is the target enzyme for treating hair baldness. Finasteride, a selective SRD5A2 inhibitor, and dutasteride, an inhibitor of both SRD5A1 and SRD5A2, are approved drugs for AGA treatment. However, the SRD5A2 isoenzyme is predominantly distributed in various organs, including the prostate, testes, and liver [87]. Thus, undesirable side effects markedly appear after drug administration, such as sexual dysfunction, testicular pain, and mental illness [88]. Natural substances, attenuating the action of SRD5A isoenzymes, seem highly probable agents for hair loss treatment with fewer adverse effects [89,90].

In this study, BB4CMU-RBO resulted in a significant down-regulation in the gene expression of both *SRD5A1* and *SRD5A2* among all the other extracts. Interestingly, BB4CMU-RBO treatment demonstrated the gene inhibition of *SRD5A2* when compared with the standard drugs: finasteride and dutasteride (Figure 3). The highest *SRD5A2* gene inhibition activity of BB4CMU-RBO may be attributed to our previous study presenting the highest amount of β-tocopherol (1.16 ± 0.01 mg/100 g crude fat), followed by γ-tocopherol (6.78 ± 0.04 mg/100 g crude fat), and α-tocopherol (15.84 ± 0.03 mg/100 g crude fat), consecutively [11]. Moreover, we also reported the binding affinity of bioactive compounds in rice bran extract using molecular docking, in which β-tocopherol, γ-tocopherol, and α-tocopherol can possess a specific binding to SRD5A2 [91]. Therefore, *SRD5A2* inhibition by the BB4CMU-RBO extract may be attributed to the structural features of tocopherols. Our data agreed with the recent studies of *Argania spinosa* [92] and *Prunus mira* [93], which displayed hair growth promotion related to the contents of tocopherols. Our findings demonstrated that BB4CMU-RBO, with its main tocopherol compounds, essential fatty acids, and mineral contents presented properties with a high potential for regenerating hair.

## 3. Materials and Methods

### 3.1. Chemicals and Reagents

The materials 2,2-Diphenyl-1-picrylhydrazyl (DPPH), 2,2′-azino-bis (ethylbenzthiazoline-6-sulfonic acid (ABTS), 3-(2-yyridyl)-5,6-diphenyl-1,2,4-triazine-4′,4′′-disulfonic acid sodium salt (ferrozine), iron (II) chloride tetrahydrate (FeCl_2_ · 4H_2_O), 6-hydroxy-2,5,7,8-tetramethylchroman-2-carboxylic acid (Trolox), ethylenediaminetetraacetic acid (EDTA), sulforhodamine B (SRB), D-glucose monohydrate, diclofenac sodium, lipopolysaccharide (LPS), arbutin, theophylline were purchased from Sigma Aldrich (St. Louis, MO, USA). Roswell Park Memorial Institute 1640 medium (RPMI-1640), Dulbecco’s Modified Eagle Medium (DMEM), Eagle’s Minimal Essential Medium (MEM), fetal bovine serum (FBS), 0.5% Trypsin-EDTA (10X), and 3-isobutyl-1-methylxanthine (IBMX) were obtained from Gibco Life Technologies (Grand Island, NY, USA). Penicillin/streptomycin solution (100×) was obtained from Capricorn Scientific GmbH (Ebsdorfergrund, Germany). The Griess reagent kit was acquired from Invitrogen, Thermo Fisher Scientific (Waltham, MA, USA). Finasteride and dutasteride were purchased from Wuhan W&Z Biotech (Wuhan, China). RedSafe™ dye was obtained from iNtRON Biotechnology (Gyeonggido, Korea). All other chemical substances were of analytical grade.

### 3.2. Plant Materials and Preparation of Sample Extraction

*Oryza sativa* cv. Bue Bang 3 CMU (BB3CMU) and Bue Bang 4 CMU (BB4CMU) were obtained from the Lanna Rice Research Center, Chiang Mai University, Chiang Mai, Thailand. Both rice cultivars were grown under the same environmental conditions for six months, between June and December 2021. The rice extracts were provided by the Pharmaceutical and Natural Products Research and Development Unit (PNPRDU), Chiang Mai University, Chiang Mai, Thailand. The sample extraction followed the scheme illustrated in Figure 4. Briefly, the rice husk was first removed from the rice seed, and then the rice bran was separated from the white rice grain. Rice bran (1000 g) was fed into the mechanical screw press to obtain the rice bran oil part 1 and its residue. Then, the rice bran residue part 1 (100 g) was further extracted using the dichloromethane extraction (2 L) for 72 h to obtain the rice bran oil part 2 and the rice bran residue part 2. The rice bran oil (RBO) in this study was a mixture of the rice bran oil parts 1 and 2. After that, the rice bran residue part 2 (100 g) was macerated with 95% ethanol (2 L) for 72 h and then filtered and concentrated by a rotary evaporator to obtain the defatted rice bran extract (DFRB). Moreover, the dried husk (100 g) was mixed with 95% ethanol (6 L) and followed with a 48-h maceration to obtain the rice husk extract (H). All stages of the extraction methods were performed at room temperature. The rice samples were labeled as follows: BB3CMU-RBO, BB3CMU-DFRB, BB3CMU-H, BB4CMU-RBO, BB4CMU-DFRB, and BB4CMU-H. Consequently, all extracts were kept at a temperature of 4 °C until used.

### 3.3. Antioxidant Activity Assays

The antioxidant potential of the samples was performed using the DPPH radical scavenging assay, the ABTS radical scavenging assay, and the iron chelating assay, as previously described [94]. Briefly, each sample was diluted in the range of 0.01–10 mg/mL. For the DPPH assay, 50 µL of the sample solution was mixed with 50 µL of DPPH working solution, then incubated in a dark at room temperature for 30 min. The absorbance of each well was determined at 517 nm using the EZ2000 microplate reader (Biochrome Ltd., Cambridge, UK). For the ABTS assay, 25 µL of the sample solution was reacted with 200 µL of the ABTS working solution, then incubated at room temperature for 10 min. The absorbance was measured at 734 nm. Trolox was used as the standard scavenging agent for the DPPH and ABTS assays. For metal chelation, 100 µL of the sample solution was reacted with 50 µL of 2 mM ferrous chloride solution and 50 µL of 5 mM ferrozine substrate. After incubating at room temperature for 30 min, the absorbance was measured at 562 nm. EDTA was used as the standard chelating agent. Blanks for every sample without free radical or heavy metal solutions were also conducted for each measured absorbance. The results were obtained from the line slope of each sample, which was divided by the line slope of the standard and expressed as mg of Trolox equivalent/g of extract and mg of EDTA equivalent/g of extract [95].

### 3.4. Cell Cultures

Mouse skin melanoma cells (B16F10; JCRB no. JCRB0202) and human fibroblast cells (JCRB no. JCRB1006.4F) were purchased from the JCRB Cell Bank (Osaka, Japan). Murine RAW 264.7 macrophage cells and human prostate cancer cells (DU-145) were purchased from the American Type Culture Collection (Rockville, MD, USA). Primary human follicle dermal papilla cells (HFDPC) were obtained from Promo Cell GmbH (Heidelberg, Germany). The B16F10 cells were grown in a MEM culture medium supplemented with 10% FBS and 1% penicillin/streptomycin. The RAW 264.7 cells were grown in a DMEM culture medium supplemented with 4500 mg/L D-glucose, 10% FBS, and 1% penicillin/streptomycin. The DU-145 cells were grown in RPMI-1640 culture medium supplemented with 10% FBS and 1% penicillin/streptomycin. The fibroblast cells were cultured in a DMEM culture medium supplemented with 10% FBS, and 1% penicillin/streptomycin. The HFDPC cells were cultured in a Growth Medium Kit supplemented with 10% FBS and 1% antibiotic-antimycotic 100× solution. All cell lines were maintained under a humidified atmosphere containing 5% CO_2_ at a temperature of 37 °C.

### 3.5. Cytotoxicity Assay

The cell viability of B16F10, RAW 264.7, DU-145, fibroblast, and HFDPC cells for determining non-toxic concentrations (above 80% cell viability) was evaluated through the colorimetric SRB assay [10]. The cells were seeded into 96-well plates at a density of 1 × 10^4^ cells/well. After 24 h, the growth medium was replaced with fresh serum-free medium containing various concentrations (0.0001–1 mg/mL) of rice extracts and standard controls on RAW 264.7, DU-145, and HFDPC cells for 24 h and on B16F10 and fibroblast cells for 48 h. Following incubation, 50% trichloroacetic acid was added to fix cells for 1 h at a temperature of 4 °C. The cells were rinsed with deionized water and air-dried. After that, cells were stained using a 0.04% SRB solution for 30 min at room temperature, and then the unbound dye was discarded by washing cells with 1% acetic acid and air-drying. The bound dye was dissolved with a 10 mM Tris base. The absorbance was measured at 515 nm using a microplate reader. The cell viability of the treated cells was compared with untreated cells as the percentage of cell viability of control.

### 3.6. Melanin Content Assay

This assay was examined on murine melanoma cells by a slight modification method [96,97]. Briefly, the B16F10 melanoma cells (2.5 × 10^5^ cells/well) were seeded in 6-well plates. After 24 h of incubation, the cells were treated with serum-free medium containing 50 µM IBMX and the rice samples or the standard controls (theophylline and arbutin) at a concentration of 0.01 mg/mL for an additional 48 h. Then, the cell pellets were harvested using 0.25% trypsin-EDTA and centrifuged at 10,000× *g* for 10 min. The intracellular melanin was dissolved in 1 N NaOH containing 10% DMSO at a temperature of 80 °C for 30 min. The amount of melanin was measured using an absorbance of 405 nm. The results were expressed as the relative percentage of melanin content of the control (untreated cells).

### 3.7. Intracellular Nitric Oxide Production

The nitric oxide production in macrophage cells was expressed as the amount of accumulated nitrite in the cell supernatant. This method was modified from a previously published method [98,99]. Briefly, the RAW 264.7 cells were plated at a density of 1 × 10^5^ cells/well in 96-well plates for 24 h. Then, the cells were induced by 1 µg/mL LPS for 24 h after pre-treatment with the rice extracts (RBO, 0.10 mg/mL; DFRB, 0.0001 mg/mL; and H, 0.10 mg/mL) or the standard diclofenac sodium (0.10 mg/mL). The supernatant was collected to react with the Griess reagent according to the manufacturer’s recommendations. The nitrite content of each sample was calculated using the standard curve of sodium nitrite.

### 3.8. RNA Extraction and Semi-Quantitative Reverse Transcription Polymerase Chain Reaction

The modulation of the *SRD5A* gene expression was observed in prostate tumor cells as previously described [10]. The DU-145 cells were used as an in vitro cell model to determine the expression of 5α-reductase for anti-hair loss applications [100,101,102] because the action of 5α-reductase was obviously evaluated in prostate cells [103]. Total RNA was isolated from the DU-145 cells treated with samples (RBO, 0.25 mg/mL; DFRB, 0.50 mg/mL; and H, 0.10 mg/mL) or SRD5A enzyme inhibitors (dutasteride and finasteride at a concentration of 0.10 mg/mL) for 24 h. The RNA extraction process was performed using the E.Z.N.A.^®^ Total RNA Kit I (Omega Bio-Tek, Norcross, GA, USA) following the manufacturer’s instructions. The RNA concentrations were determined using the Qubit™ 4 fluorometer and Qubit™ RNA HS Assay Kit (Invitrogen, Carlsbad, CA, USA). Complementary DNA (cDNA) was synthesized using the RT-PCR Quick Master Mix (Toyobo, Osaka, Japan) as the template and the following primers for amplification: *SRD5A1* (F: 5′-AGCCATTGTGCAGTGTATGC-3′ and R: 5′-AGCCTCCCCTTGGTATTTTG-3′), *SRD5A2* (F: 5′-TGAATACCCTGATGGGTGG-3′ and R: 5′-CAAGCCACCTTGTGGAATC-3′), *SRD5A3* (F: 5′-TCCTTCTTTGCCCAAACATC-3′ and R: 5′-TCCTTCTTT-GCCCAAACATC-3′), and *GAPDH* (F: 5′-GGAAGGTGAAGGTCGGAGTC-3′ and R: 5′-CTCAGCCTTGACGGTGCCATG-3′). *GAPDH* was used to normalize the expression of the target genes. The PCR product was separated through electrophoresis on 1% agarose gel and stained using RedSafe™ dye. The DNA bands were visualized on a gel documentation system (Gel Doc™ EZ, Bio-Rad, Hercules, CA, USA) and Image Lab™ Software (Bio-Rad, Hercules, CA, USA).

### 3.9. Statistical Analysis

All experiments were performed in triplicate and reported as means ± standard deviation (SD). All data were analyzed using the one-way analysis of variance (ANOVA) test along with the LSD’s post hoc test using SPSS 23.0 Software (SPSS Inc., Chicago, IL, USA). The significant difference was defined as a *p*-value *<* 0.05.

## 4. Conclusions

Rice bran and rice husk are the most common agricultural waste with deteriorated environmental effects. Our previous studies illustrated the abundant constituents of rice by-products extracts from numerous Thai rice varieties. Consecutively, in this study, BB3CMU and BB4CMU were selected for biological testing using cell-based assays to identify a potential for hair nourishing effects and for restoring hair. Impressively, rice bran oil from BB4CMU showed ameliorable effects on NO-induced melanin production, which was attributed to the mineral content and free fatty acid profiles. According to the tocopherols-enrich rice bran oil fraction, the BB4CMU-RBO extract showed promising *SRD5A2* down-regulation. Possibly, BB4CMU-RBO is to be regarded as a re-pigmenting agent in graying hair and an effective treatment for androgenetic hair loss. Furthermore, BB4CMU-RBO could be used as a natural alternative treatment and a substitution for conventional drugs that possess strong side effects. Overall, the results in this study illustrated the pharmaceutical potential of plant resources for restoring hair in terms of hair growth and hair re-pigmentation.

## Figures and Tables

**Figure 1 plants-12-00653-f001:**
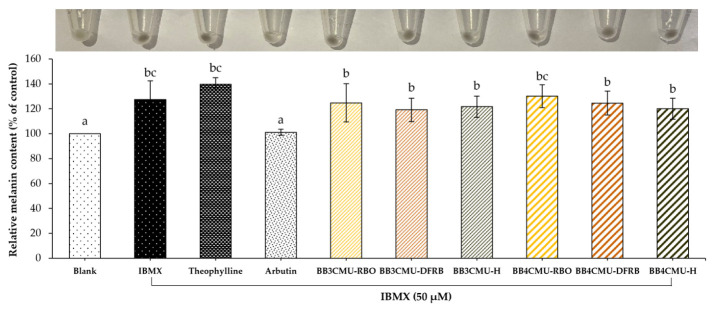
Effects of *Oryza sativa* L. cv. BB3CMU and BB4CMU extracts at 0.01 mg/mL on melanin content in B16F10 melanoma cells stimulated with 50 µM 3-isobutyl-1-methylxanthin (IBMX) for 48 h. Theophylline and arbutin (0.01 mg/mL) were used as standard controls. Cells cultured in the medium without treatment refer to the blank. Different letters within each treatment indicate significant differences (*p <* 0.05).

**Figure 2 plants-12-00653-f002:**
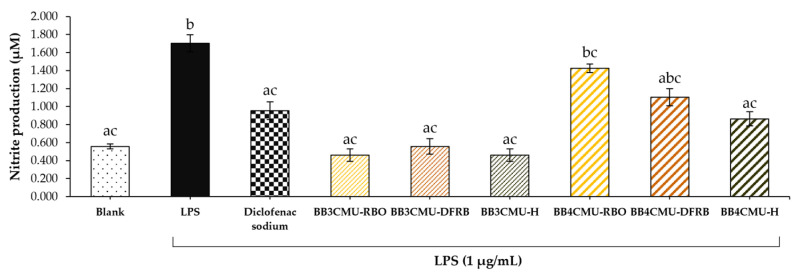
Effects of *Oryza sativa* L. cv. BB3CMU and BB4CMU extracts (RBO, 0.10 mg/mL; DFRB, 0.0001 mg/mL; H, 0.10 mg/mL) on nitrite content in RAW 264.7 macrophages induced with 1 µg/mL lipopolysaccharide (LPS) for 24 h. Diclofenac sodium (0.10 mg/mL) was used as standard control. Cells cultured in the medium without treatment refer to the blank. Different letters within each treatment indicate significant differences (*p <* 0.05).

**Figure 3 plants-12-00653-f003:**
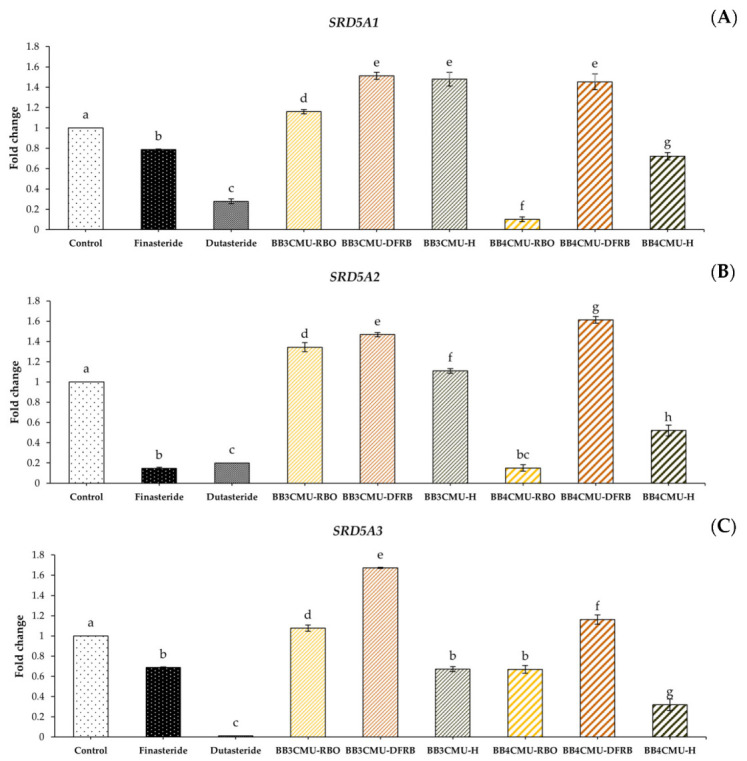
Effects of *Oryza sativa* L. cv. BB3CMU and BB4CMU extracts (RBO, 0.25 mg/mL; DFRB, 0.50 mg/mL; H, 0.10 mg/mL) on mRNA expression of (**A**) *SRD5A1*; (**B**) *SRD5A2*; (**C**) *SRD5A3* in DU-145 prostate cancer cells after 24 h of treatment. Finasteride and dutasteride (0.10 mg/mL) were used as standard controls. Different letters within each treatment indicate significant differences (*p <* 0.05).

**Figure 4 plants-12-00653-f004:**
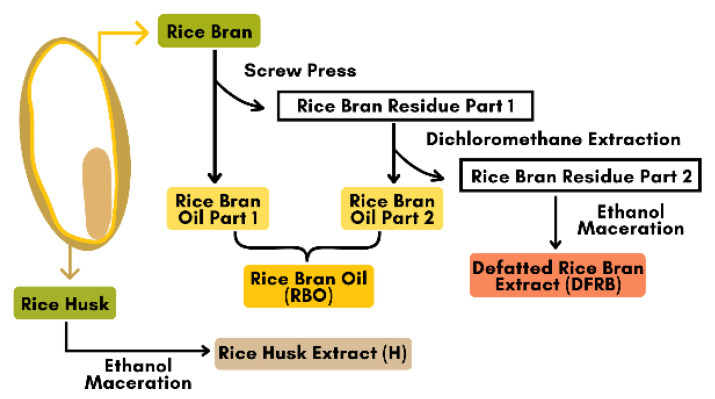
Scheme of rice by-products extraction from *Oryza sativa* L. cv. Bue Bang 3 CMU (BB3CMU) and Bue Bang 4 CMU (BB4CMU).

**Table 1 plants-12-00653-t001:** Antioxidant potential of *Oryza sativa* L. cv. BB3CMU and BB4CMU extracts.

Extracts	DPPH Radical Scavenging Activity (mg TE/g Extract)	ABTS Radical Scavenging Activity (mg TE/g Extract)	Iron Chelating Activity (mg EDTAE/g Extract)
BB3CMU−RBO	70.14 ± 0.97 ^a^	5.45 ± 0.28 ^a^	28.79 ± 5.63 ^a^
BB3CMU−DFRB	147.65 ± 2.05 ^b^	51.83 ± 2.67 ^b^	65.47 ± 12.79 ^a^
BB3CMU−H	334.70 ± 4.64 ^c^	196.13 ± 10.09 ^c^	283.22 ± 55.35 ^b^
BB4CMU−RBO	37.05 ± 0.51 ^d^	3.84 ± 0.20 ^a^	24.28 ± 4.74 ^a^
BB4CMU−DFRB	117.84 ± 1.63 ^e^	47.03 ± 2.42 ^b^	71.78 ± 14.03 ^a^
BB4CMU−H	198.04 ± 2.75 ^f^	164.10 ± 8.45 ^d^	61.12 ± 11.94 ^a^

BB3CMU-RBO, rice bran oil of *Oryza sativa* L. cv. Bue Bang 3 CMU; BB3CMU-DFRB, defatted rice bran extract of *Oryza sativa* L. cv. Bue Bang 3 CMU; BB3CMU-H, husk extract of *Oryza sativa* L. cv. Bue Bang 3 CMU; BB4CMU-RBO, rice bran oil of *Oryza sativa* L. cv. Bue Bang 4 CMU; BB4CMU-DFRB, defatted rice bran extract of *Oryza sativa* L. cv. Bue Bang 4 CMU; BB4CMU-H, husk extract of *Oryza sativa* L. cv. Bue Bang 4 CMU; TE, Trolox equivalent; EDTAE, EDTA equivalent. Different letters within each treatment indicate significant difference at *p*-value < 0.05.

## Data Availability

Not applicable.

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
