# Peer review of "Regulatory Effects of Thai Rice By-Product Extracts from Oryza sativa L. cv. Bue Bang 3 CMU and Bue Bang 4 CMU on Melanin Production, Nitric Oxide Secretion, and Steroid 5α-Reductase Inhibition"

_plants, 2023, doi:10.3390/plants12030653_

Round 1
Reviewer 1 Report
The manuscript entitled "Regulatory Effects of Thai Rice By-Product Extracts from 2 Oryza sativa L. cv. Bue Bang 3 CMU and Bue Bang 4 CMU on 3 Melanin Production, Nitric Oxide Secretion and Steroid 4 5α-Reductase Inhibition" explores the biological activity of rice by products as hair care ingredients.
The work is interesting and entails an extensive review of the literature. However, the English needs major revision and is almost incomprehensible.
The work slightly refers the extraction process and conditions. If this was already published in previous work, should be referred in the text. If not, the protocol should be further explained and contain further information regarding: ratios, time and temperature of extraction and yield. Also, there are no chemical characterization of the extracts regarding carbohydrates compositions, water content, ashes, etc.
Regarding the cell lines used: Raw 264.7, B16F10 and DU-145, i can't find the relevance for the use of human prostate cancer cell lines in the present study. Further, the citotoxicity of the extract should be evaluated in keratinocytes and fibroblasts to infer the citotoxicity in healthy scalp.
For cosmetic applications it is of most importance to evaluate the potential sensitizer potential of the ingredients through DPRA test and/or HClat tests.
Some conclusions such as: "This result suggested that the melanogenesis stimulatingability of BB4CMU-RBO may be contributed to iron, zinc, and unsaturated fatty ac-ids" are extrapolation and need fruther fundamentation.
Author Response
Dear Reviewer 1,
The revised manuscript has been thoroughly checked and corrected (point-by-point response to each comment).
Please see the attachment.
Sincerely,
Warintorn et al.

Reviewer 3 Report
The manuscript “Regulatory Effects of Thai Rice By-Product Extracts from Oryza sativa L. cv. Bue Bang 3 CMU and Bue Bang 4 CMU on Melanin Production, Nitric Oxide Secretion and Steroid 5α-Reductase Inhibition” is interesting and brings new information on antioxidant properties, melanin production, nitric oxide production, and inhibitory effects on steroid 5α-reductase isoenzymes, of three different rice by-products of BB3CMU and BB4CMU as botanical therapeutic agents. Hence, I believe that results provided are important, methodology adequate and therefore this paper can be published in Plants Journal in present form.
Author Response
Dear Reviewer 2,
The revised manuscript has been thoroughly checked and corrected.
Please see the attachment.
Sincerely,
Warintorn et al.

Round 2
Reviewer 1 Report
The manuscript entitled "Regulatory Effects of Thai Rice By-Product Extracts from 2 Oryza sativa L. cv. Bue Bang 3 CMU and Bue Bang 4 CMU on Melanin Production, Nitric Oxide Secretion and Steroid 5α-Reductase Inhibition" was successfully reviewed by the authors and alterations made improved the final work.
I would just like to acknowledge the fact that the images, especially the the graphs have very low resolution and need to be more perceptible.